# Prediction Model of Aluminized Coating Thicknesses Based on Monte Carlo Simulation by X-ray Fluorescence

Zhuoyue Li, Cheng Wang *, Haijuan Ju, Xiangrong Li, Yi Qu and Jiabo Yu

Fundamentals Department, Air Force Engineering University, Xi'an 710051, China; lz_980512@163.com (Z.L.); jhjcumtgx@163.com (H.J.); lixiangrong0925@126.com (X.L.); strsky778@163.com (Y.Q.); b2283216046@163.com (J.Y.)
* Correspondence: valid_01@163.com

**Abstract:** An aluminized coating can improve the high-temperature oxidation resistance of turbine blades, but the inter-diffusion of elements renders the coating's thickness difficult to achieve in non-destructive testing. As a typical method for coating thickness inspection, X-ray fluorescence mainly includes the fundamental parameter method and the empirical coefficient method. The fundamental parameter method has low accuracy for such complex coatings, while it is difficult to provide sufficient reference samples for the empirical coefficient method. To achieve accurate non-destructive testing of aluminized coating thickness, we analyzed the coating system of aluminized blades, simulated the spectra of reference samples using the open-source software XMI-MSIM, established the mapping between elemental spectral intensity and coating thickness based on partial least squares and back-propagation neural networks, and validated the model with actual samples. The experimental results show that the model's prediction error based on the back-propagation neural network is 4.45% for the Al-rich layer and 16.89% for the Al-poor layer. Therefore, the model is more suitable for predicting aluminized coating thickness. Furthermore, the Monte Carlo simulation method can provide a new way of thinking for materials that have difficulty in fabricating reference samples.

**Keywords:** X-ray fluorescence; aluminized coating; Monte Carlo simulation; turbine blades; backward propagation neural network; XMI-MSIM

## 1. Introduction

Airline flight safety is a primary concern for countries worldwide, and airline crashes cause significant economic losses and are also devastating to the families of victims. A reliable engine is key to protecting the aircraft for safe flights. For Turbine blades as the engine power-energy transfer exchanger, frequently in high-temperature, high-pressure states, this state on the turbine blade's material performance put forward higher requirements [1–4]. Generally, the turbine blade substrate is made of a nickel-based high-temperature alloy with an aluminizing treatment on the surface to prevent high-temperature oxidation [5]. However, the harsh service environment damages the aluminum coating on the blade's surface, affecting the high-temperature oxidation resistance of the blade [1,6]. Therefore, it is necessary to monitor turbine blade coating conditions. The commonly used method for checking coating thickness is the cross-sectional metallographic method [7], which is straightforward and effective but will damage the blade. The method is costly and is usually performed on a random sample. Hence, developing a non-destructive testing (NDT) method for the Al coating on turbine blade surfaces is urgently required.

Various NDT methods have been tested with an aim to achieve non-destructive testing of the thickness of diffusion coatings on turbine surfaces, but each method has its limitations. The ultrasonic method is generally used for coatings with obvious physical boundaries, while the physical boundaries of diffusion coatings are not obvious. The eddy current

method is generally used for non-metallic coatings on non-magnetic metal substrates, while both diffusion coatings and their substrates are metals. The magnetic thickness measurement method is generally used for non-magnetic coatings on magnetic substrates, and diffusion coatings are also not applicable [8].

As an elemental analysis technique, X-ray fluorescence (XRF) is not only used in element identification and element quantification but also widely used in coating thickness measurement [9–13]. The current XRF coating thickness measurement is mainly based on the fundamental parameter method without standard samples and the empirical coefficient method with standard samples [14–16]. For gold and silver coatings on the surface of precious objects, Brocchieri [10,17] used the empirical coefficient method combined with partial least squares regression for coating thickness prediction, obtaining relatively consistent results with the expected thickness. Takahara [18] introduced the advantages of the fundamental parameter method for thickness measurement when no standard samples are available and explained the details of the fundamental parameter method for thickness measurement using an ITO film sample as an example. However, for turbine blades in service with a large variety of elements and mutual diffusion between coating and substrate elements [19], the fundamental parameter method is not accurate enough for calculation, and it is not possible to provide sufficient reference samples for the empirical coefficient method. Therefore, a Monte Carlo simulation is considered to provide the spectra of the reference samples.

Monte Carlo simulation uses a statistical method to simulate the process of photon-matter interaction. Schoonjans [20–22] developed the open-source software XRMC and XMI-MSIM for EDXRF Monte Carlo simulations, which can be performed by setting parameters such as excitation source, instrument geometry layout, detector, and sample composition. Giurlani [11,12] used this method to establish standard curves for determining the thickness of single and multilayer metal coatings and obtained significantly better results than the fundamental parameter method. Trojek [23] used iterative Monte Carlo simulations to determine the copper alloy composition and the thickness of the cover layer and verified the efficiency and robustness of the method. A series of studies applying Monte Carlo simulation methods in archaeology is also available [24–26].

In this study, in order to achieve a non-destructive detection of aluminized coating thickness on the in-service turbine blade's surface, we first analyzed the blade cross-section to determine the layering and composition of the coating, obtained XRF spectra at different thicknesses using Monte Carlo simulation, established a coating thickness prediction model using chemometric methods, and finally verified the accuracy of the model by testing the XRF spectra on the blade's surface.

## 2. Materials and Methods

### 2.1. Principle of XRF Thickness Measurement

The basic principle of XRF elemental analysis is as follows: primary X-rays excite the elements in the sample to produce XRF, identify the elemental species based on the difference in XRF energy of different elements, and determine the content of the elements in the sample based on XRF intensity [16]. Traditional methods for mapping relationships between XRF intensity and elemental content include the empirical coefficient method [14] and the fundamental parameter method [15]. The empirical coefficient method establishes a calibration curve by measuring a series of reference samples. The fundamental parameter method requires knowing the exact parameters of each element and determining the geometric factor of the instrument by testing the spectra of several pure elements to enable a broader range of standardless measurements [27]. There are three methods of determining coating thickness by XRF as follows [28]:

1. Emission method: The research object of the emission method is the coating element. The thicker the coating, the greater the XRF intensity of the coating element.
2. Absorption method: The research object of the absorption method is the substrate element. The thicker the coating, the more the XRF of the substrate element is absorbed, and the smaller the intensity.
3. Relative method: The relative method calculates the ratio of the XRF intensity of the coating element and the substrate element to determine the thickness [16]. Since the method uses relative values to calculate thickness, variations in measurement conditions have little effect on the calculation error.

It is difficult for aluminized coatings on turbine blade surfaces to obtain accurate results simply using the above thickness measurement methods due to the inter-diffusion between elements. We need to explore the mapping between the XRF intensity of multiple elements and coating thickness by measuring a series of reference samples of aluminized coatings with different thicknesses. However, such reference samples are difficult to fabricate. The Monte Carlo method can simulate interactions, including scattering effects, photoelectric effects, and Auger electrons, considering the entire spectrum of analytes. Therefore, we use the Monte Carlo method to simulate a series of reference samples to establish the mapping relationship between XRF intensity and coating thickness.

### 2.2. Experimental Procedure

The experiments mainly include characterization, simulation, and validation processes, and mathematical modeling is included in the simulation process. The flow chart of the research process is summarized in Figure 1.

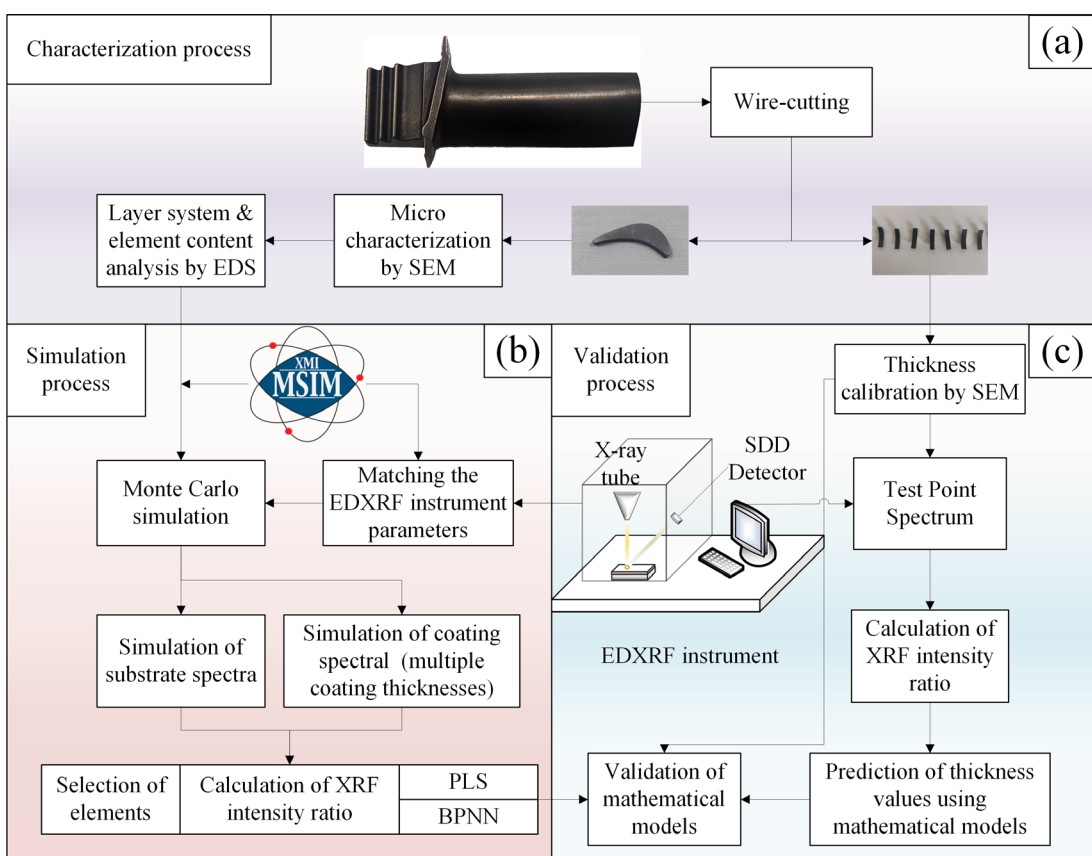

**Figure 1.** Flow chart of the characterization, simulation, and validation process. (**a**) Characterization process; (**b**) Simulation process; (**c**) Validation progress.

Efforts in the characterization process include sample preparation, micromorphological characterization, analysis of the coating system, and elemental content. A type of in-service turbine blade provided by Xiangyang Hangtai Power Machine Factory was used for characterization experiments. The blade substrate is DZ422B nickel-based superalloy, protected by an aluminized layer on the surface. It was the first to cut into eight pieces equidistantly along the blade body of the turbine blade using an EDM wire cutting machine (DK7725H, ZHCF, Suzhou, China). The cross-section of the pieces was then sanded and polished with graded grit sandpapers to achieve a roughness where the surface coating could be clearly observed. Prior to SEM (VEGA-3 XMU, TESCAN, Brno, Czech Republic) observation, the slices were cleaned using an ultrasonic cleaner (KM-410C, KJM, Guangzhou, China), dried, and wiped on the surface with alcohol to avoid contaminants from affecting characterization results. The slices were placed on the sample stage with the smooth section facing upward through the conductive adhesive, the acceleration voltage was set to 20 kV, and the current was set to 1 nA. The electron beam was switched on, and the magnification was $1000\times$. The backscatter detector was selected to make the interface contrast between the coating and the substrate more obvious. After the analysis area was selected, a surface scan of the area was performed using EDS (AZtecOne & x-act, Oxford, London, UK) to obtain the overall distribution of various elements, and the EDS worked with the same parameters as SEM. A line scan was performed along the coating depth direction to obtain the trend of different elements in that direction. Finally, a point scan was performed for the areas with apparent boundaries to obtain the elemental content values of each area.

The simulation process is based on the characterization process and the determination of the EDXRF instrument (XAU-4CS, YL, Suzhou, China) parameters. Monte Carlo simulations were performed using the open-source software XMI-MSIM developed by Schoonjans T. [21]. The software can automatically generate the simulated energy spectrum of primary X-rays by inputting parameters, adjusting parameters to set the geometric layout between excitation source-sample-detector, and inputting elemental content to set the sample's composition, including the air layer. The main parameter settings of the simulation process are shown in Table 1. The sample composition was set according to the coating system and elemental content obtained during the characterization. The remaining parameters of the Monte Carlo simulation were set according to the excitation conditions, geometric layout, and detector type of the EDXRF instrument. The coating system studied in this paper is divided into three layers. The first layer is set up with 13 different thickness values in the range of 0-60 μm at every 5 μm step. The second layer is set up with 11 different thickness values in the range of 0–40 μm at every 4 μm step. The third layer is set up with 4 different thickness values of 0 μm, 8 μm, 15 μm, and 22 μm. The spectra and XRF intensities of the main elements were recorded in detail for each simulation to obtain $13 \times 11 \times 4 = 572$ sets of data. Before establishing the mathematical model of XRF intensity of the major elements versus coating thickness, the ratio ($R$) of the XRF intensity value ($I_c$) obtained from the simulation of the major elements in the coating to the XRF intensity value ($I_s$) obtained from the simulation of the corresponding elements in the substrate was calculated to mitigate the deviation formed during the testing of the instrument and simulation software. The data obtained from the simulation were randomly divided in the ratio of training set: test set = 4:1, and then the more mature chemometric methods, such as partial least squares (PLS) and backward propagation neural networks (BPNN), were used to model the data and compare their goodness of fit. The mathematical methods mentioned above have been widely used in XRF [29–33].

**Table 1.** Parameters for Monte Carlo simulation.

| Project | Segmentation | Parameter | Density (g/cm³) | Thickness (cm) |
|---------|-------------|-----------|-----------------|----------------|
| General | Number photons per interval | 10,000 | | |
| | Number of photons per discrete line | 100,000 | | |
| | Number of interactions per trajectory | 4 | | |
| Geometry | Sample-source distance (cm) | 2 | | |
| | Primary X-ray incidence angle (°) | 90 | | |
| | X-ray fluorescence emission angle (°) | 45 | | |
| | Active detector area (cm²) | 0.0531 | | |
| Excitation | Tube voltage (kV) | 40 | | |
| | Tube current (mA) | 1.00 | | |
| | Anode | W | 19.3 | 0.0002 |
| | window | Be | 1.848 | 0.0125 |
| Composition | Air, Dry | C, N, O, Ar | 0.001205 | 2 |
| | Al-rich layer | | 5.24 | 0~0.0022 |
| | Al-poor layer | Reference | 6.05 | 0~0.0040 |
| | IDZ layer | Table 2 | 6.83 | 0~0.0060 |
| | Substrate | | 7.28 | 1 |
| Detection | Detector type | SDD | | |
| | Number of spectrum channels | 4096 | | |
| | Active detector area (cm²) | 0.25 | | |

**Table 2.** Elemental composition of coatings and substrate (wt.%).

| Element | C | Al | Cl | K | Ti | Cr | Co | Ni | Nb | Yb | Hf | W |
|---------|-----|------|------|------|------|-------|------|-------|------|------|------|-------|
| Al-rich layer | 7.64 | 20.52 | 0.33 | 0.16 | 0.42 | 3.54 | 8.20 | 53.68 | 0 | 0 | 0 | 5.51 |
| Al-poor layer | 6.47 | 11.47 | 0.25 | 0 | 1.71 | 6.36 | 9.51 | 60.35 | 0.38 | 0 | 0 | 3.51 |
| IDZ layer | 7.16 | 6.21 | 0.21 | 0 | 2.80 | 11.21 | 9.39 | 41.57 | 1.40 | 0 | 0 | 20.06 |
| Substrate | 5.37 | 2.81 | 0.22 | 0.09 | 2.48 | 8.25 | 9.48 | 56.14 | 1.01 | 2.07 | 1.47 | 10.61 |

The validation process is based on the slices obtained from the characterization process and the mathematical model obtained from the simulation process to verify the model's accuracy. The cross-section of the slices was first observed using an SEM to calibrate the coating's thickness. Although the slices belong to the same turbine blade, the different locations are subjected to different impacts and loads in high-temperature environments; therefore, their coating thicknesses also differ. The EDXRF instrument was then used to test the XRF spectra of the slices, and the instrument parameters were consistent with those set for the simulation process. The ratio of the XRF intensity of the main elements to the corresponding elements of the substrate is calculated and substituted into the mathematical model established by the simulation process to predict the coating's thickness. Finally, the predicted thickness was compared with the calibrated thickness to verify the model's accuracy in the actual test.

The dependence between the individual coating thicknesses was not considered during the experiments; thus, some of the simulated conditions would not occur on

the actual blade's surface. However, these simulated data are still consistent with the variation pattern of XRF intensity in the coating and have a negligible impact on the mathematical model.

## 3. Results and Discussion

### 3.1. Characterization of the Coating System

Prior to performing Monte Carlo simulations, the sample's exact elemental compositions and distributions need to be clearly known. For this purpose, the turbine blade cross-section was characterized using SEM, and the mass fraction of each coating and substrate element was analyzed by EDS. The analysis result is shown in Figure 2.

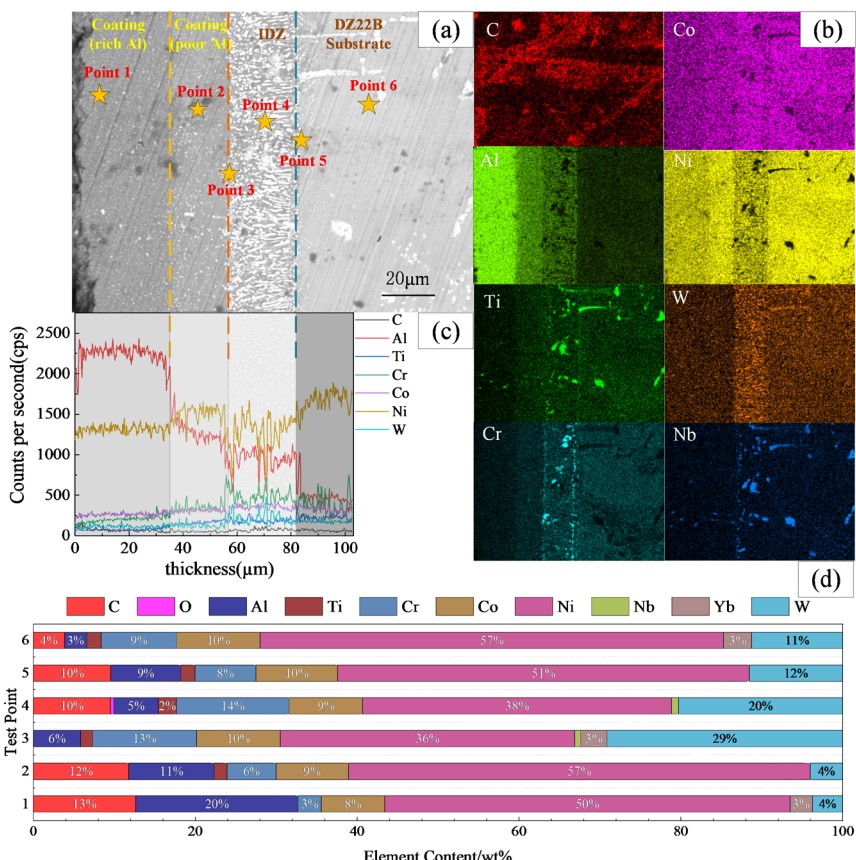

**Figure 2.** Characterization of turbine blade cross-section. (**a**) SEM morphology under backscattering. (**b**) EDS mapping of (**a**); (**c**) distribution of elements after line scan along the depth direction; (**d**) mass fraction of elements from the test points in (**a**).

In Figure 2a, it can be observed that the aluminized coating on the in-service turbine blade surface is divided into three layers. The darker color of the outermost layer indicates the higher aluminum content of this layer (under backscattering irradiation, the contrast between light and dark reflects the atomic number relationship of the elements: the darker the atomic number, the smaller and brighter the opposite is). The outermost layer corresponds to the 0–35 μm segment in Figure 2c, which we call the Al-rich layer. The second outer layer is slightly brighter than the outermost layer, corresponding to the 36–57 μm section in Figure 2c, dominated by the nickel-rich NiAl phase, which we call the Al-poor layer. For the third layer, brighter and darker areas are interspersed. This layer is formed by the mutual diffusion of the alloying elements in the substrate and the coating elements, which corresponds to the 58–82 μm segment in Figure 2c. The mass fraction of elements in this layer fluctuates widely, and we call it the interdiffusion zone (IDZ) layer. The last one is the uncoated DZ22B substrate. According to Figure 2b,d, the trends of elemental contents

are shown from the surface and point perspectives, respectively. Al content decreases along the depth direction, while the W content is the highest in the IDZ layer, Ni content is lowest in the IDZ layer, and Ti, Cr, and Nb contents are lower and not uniformly distributed. Co content has no significant changes in the entire system. Overall, the elemental mass fraction of each coating layer is relatively stable, and the average elemental mass fraction of each coating area can be taken as the composition value of this layer and is inputed into XMI-MSIM for simulation. The composition of each layer is shown in Table 2.

### 3.2. Monte Carlo Simulation

The EDXRF simulations were performed using the sample composition information obtained in Section 3.1 and the parameter settings in Table 1. The Al-rich layer, Al-poor layer, and IDZ layer were changed in equal steps within their corresponding ranges of 0–60 μm, 0–40 μm, and 0–22 μm, and the substrate was set to 1cm, which could be considered as infinite thickness in XRF simulation. For each thickness group, XRF spectra were obtained separately. The areas of the spectral peaks in the energy range of Ni Kα, Cr Kα, Co Kα, W Lα, Ti Kα, Nb Kα, Yb Lα, and Hf Lβ were calculated to obtain the XRF intensity values of the corresponding spectral lines. Compared with the XRF intensity values of the substrate's corresponding elements, obtain the XRF intensity relative values, as shown in Figure 3, where the three axes correspond to three coating thicknesses. The size of the sphere indicates the relative value of elemental XRF intensity magnitude.

In Figure 3, the XRF intensity of Ni and Co elements is the smallest, and the XRF intensity of Cr and W elements is the largest when the IDZ layer is the thickest, and the thicknesses of Al-rich and Al-poor layers are 0 μm. When the thickness of the Al-rich and Al-poor layers gradually increases, the XRF intensity of the Ni element gradually increases, and the increasing trend gradually slows down until it no longer increases. Due to the high mass fraction of Ni in the sample and the coating reaching a certain thickness, the XRF of the Ni element reaches the maximum information depth. At this time, the XRF intensity no longer changes as thickness increases. Therefore, when using the XRF intensity of the Ni element to determine the thickness of the coating, only coatings with thin thickness can be identified, and when the maximum information depth is reached, only a range of coating thicknesses can be determined. The decreasing trend of the Cr element's XRF intensity in the direction of increasing thickness of the Al-poor layer is smaller than that of the Al-rich layer, and the maximum information depth is also reached up to a certain thickness.

The XRF intensity values of Co, W, and Ti show different trends as the thickness of the Al-poor and Al-rich layers increases. However, all reach the maximum information depth when thickness increases to a certain level. In addition, since the mass fraction of Co varies less throughout the coating system, the variation in XRF intensity values for different thickness groups is also tiny. This situation is undoubtedly a challenge for the prediction of coating thickness. Fortunately, the "insignificant" (small mass fraction) Nb element has a clear pattern of variation with coating thickness and does not reach the maximum information depth. The main reason for this situation is probably the sizeable atomic number of the Nb element. Thus, the higher energy of the excited generated XRF, which can penetrate a thicker coating, has not reached the maximum information depth. For Yb and Hf elements, both show almost the same trend of XRF intensity with coating thickness, resulting from the fact that these two elements are only present in the substrate and have not inter-diffused with the coating elements. However, the mass fraction of the two elements in the substrate is so tiny that the count rates of the corresponding spectral lines are too low when the coating is thick. The resolution of EDXRF would be insufficient to calculate the spectral peak areas effectively, and significant statistical errors would seriously affect the calculation's results; thus, these two elements would not be available as selected elements for modeling and validation.

Combined with the above analysis, the XRF intensity values of the remaining six elements can be used as input variables to build the thickness prediction model, with the exception of Yb and Hf elements, which are difficult to detect.

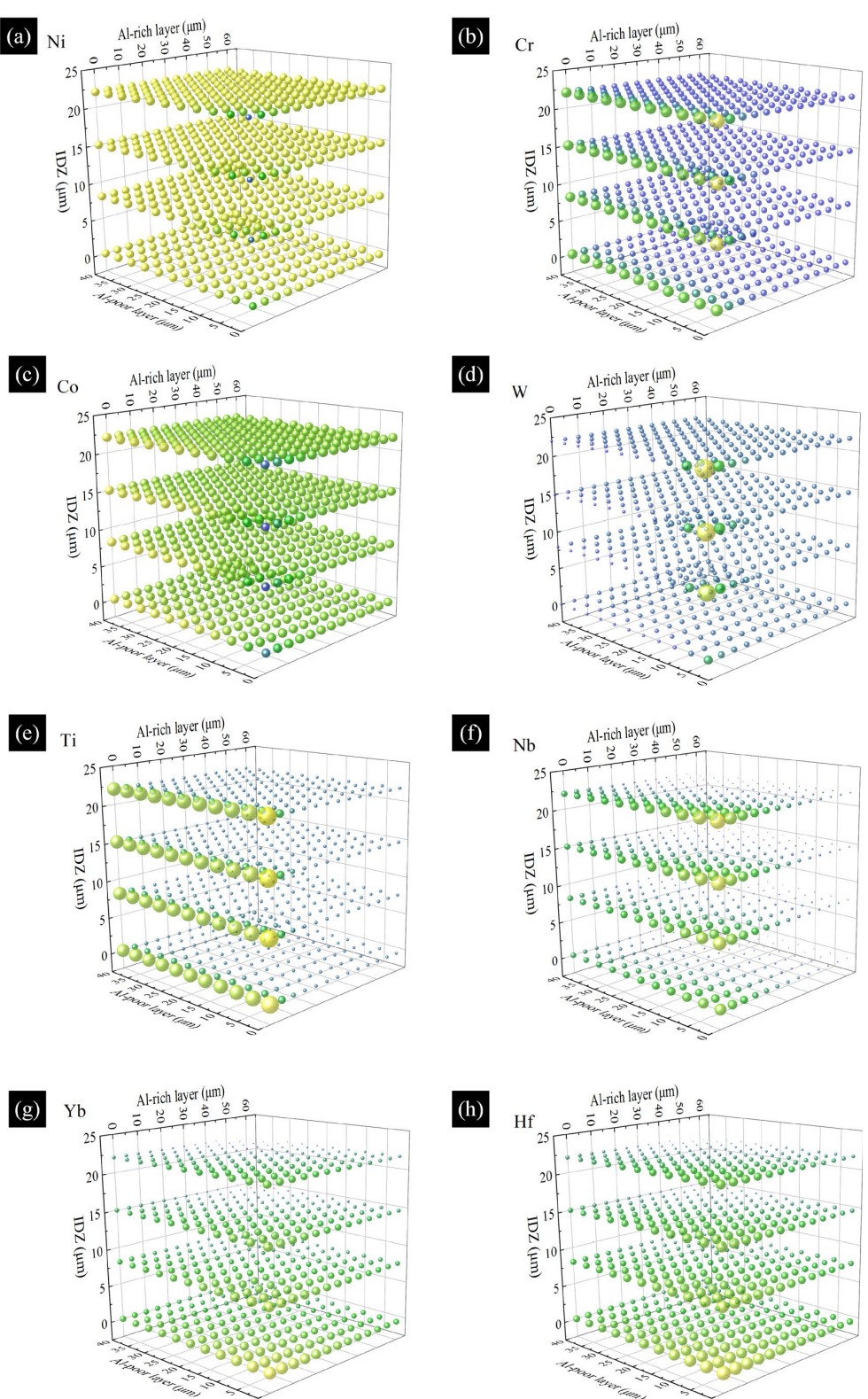

**Figure 3.** Relative values of XRF intensity as elements at different thickness combinations. (**a–h**) corresponds to Ni, Cr, Co, W, Ti, Nb, Yb, and Hf elements.

### 3.3. Thickness Prediction Modeling

Based on the Monte Carlo simulation results, suitable elements were selected, and appropriate chemometric methods were used to build the thickness prediction model.

### 3.3.1. Thickness Prediction Model based on PLS

PLS is a hidden variable method for solving the fundamental relationship between two matrices, combining the advantages of multiple linear regression analysis, typical correlation analysis, and principal component analysis. Among the 572 sets obtained from the simulation, the XRF intensity values of six elements Ni, Cr, Co, W, Ti, and Nb were used as input variables to build a 572 × 6 matrix (X). The thickness of the three coatings was used as output variables to build a 572 × 3 matrix (Y). The input variables were normalized, and then a ten-fold cross-validation method was used to find the number of PLS components that minimized the mean squared error (MSE).

Figure 4 represents the relationship between the number of PLS components and MSE, and it was observed that MSE was minimized when the number of PLS components is four. After adding additional PLS components, MSE increases instead. After determining the number of PLS components, the 572 sets of data were divided in the proportion of the training set: test set = 4:1, and 457 sets of data were obtained for building the coating thickness prediction model, and the remaining 115 sets of data were used for testing the model accuracy. The program obtained the relationship between the output variable $Y_{\text{train}547 \times 3}$ and the normalized input variable $X_{\text{train}547 \times 6}$, obtained by the program shown in Equation (1).

$$
\begin{aligned}
y_{\text{Al−rich}} &= -0.7919x_{\text{Ni}} - 3.9057x_{\text{Cr}} - 13.8979x_{\text{Co}} - 7.3017x_{\text{W}} + 9.7004x_{\text{Ti}} - 20.9559x_{\text{Nb}} + 30.0109 \\
y_{\text{Al−poor}} &= -1.2328x_{\text{Ni}} + 6.8520x_{\text{Cr}} + 18.7438x_{\text{Co}} + 13.0860x_{\text{W}} - 5.8353x_{\text{Ti}} - 5.9412x_{\text{Nb}} + 19.8074 \\
y_{\text{IDZ}} &= -0.2143x_{\text{Ni}} + 1.1263x_{\text{Cr}} + 4.6451x_{\text{Co}} + 3.8509x_{\text{W}} - 3.0958x_{\text{Ti}} + 1.9016x_{\text{Nb}} + 11.0875
\end{aligned}
\tag{1}
$$

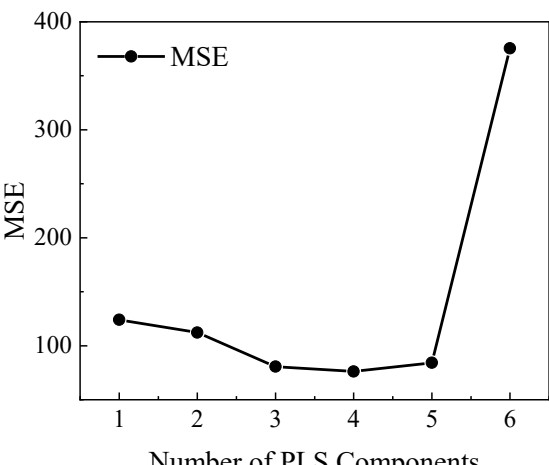

**Figure 4.** Variation of MSE with the number of PLS components.

### 3.3.2. Thickness Prediction Model Based on BPNN

BPNN is an algorithm that trains a multilayer feedforward network by error back-propagation to reveal the mapping relationship between input and output. The hidden layer between the input and output layers plays a crucial role in model building. The activation function, which connects these three basic units, is the core of BPNN in dealing with nonlinear problems. Therefore, selecting the activation function and the number of nodes of the hidden layers is crucial for obtaining a model with an excellent fitting effect and high accuracy. The same data set partitioning as in Section 3.3.1 was used in this section, with 457 data sets used to train the model and 115 data sets used to test the accuracy of the model. Based on experience, by trying different parameters such as activation function, number of hidden layers, number of hidden layer nodes, and learning rate, the best combination of parameters was obtained, as shown in Table 3. The BPNN topology diagram is shown in Figure 5.

**Table 3.** BPNN parameter settings.

| Parameters | Values |
|---|---|
| Activation function | tanh |
| Hidden layer sizes | (15, 10) |
| Learning rate | 0.02 |
| Max iteration | 5000 |

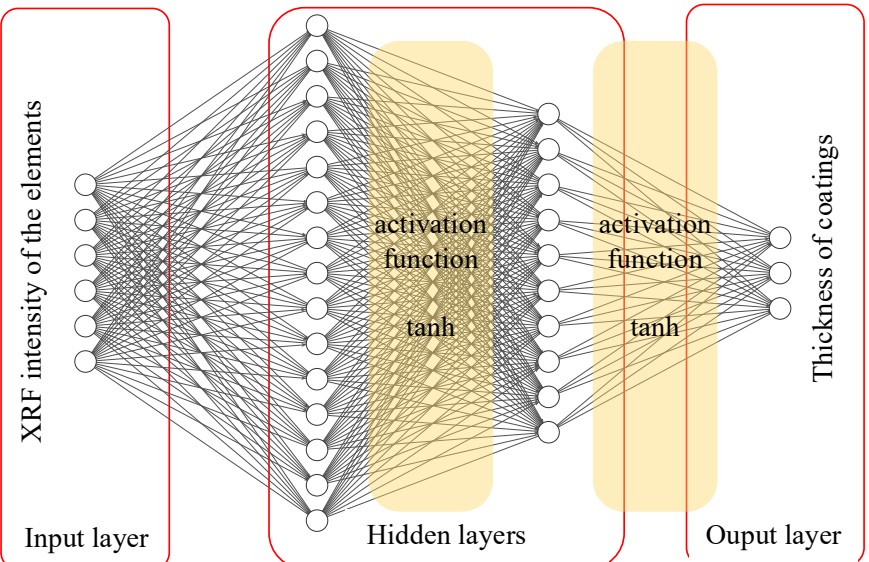

**Figure 5.** Topology of BPNN coating thickness prediction model.

### 3.3.3. Comparison of Model Results

The Al-rich and Al-poor layers will be able to form protective $Al_2O_3$ to improve the high-temperature oxidation resistance of turbine blades when exposed to air. In contrast, the IDZ layer contains less Al and cannot form adequate protection for turbine blades. In addition, for the XRF determination of the coating thickness method, the more inner layers there are, the greater the error in predicting the thickness due to the error propagation effect. Therefore, in this study, the predicted results only show the thickness of the Al-rich and Al-poor layers.

Based on the coating thickness prediction models developed in the previous two sections, the prediction results for the test set are shown in Figure 6. From the four prediction results, the best result is the prediction of Al-rich layer thickness by BPNN (Figure 6a), and the prediction of the middle part is very close to the actual thickness except for the relatively large deviation of the prediction at the two ends of the thickness range. The main reason for this marginal effect is divided into two aspects. For the thinner coatings, the XRF intensity values obtained from the tests showed little difference from the substrate, and the background noise and statistical errors would significantly impact the prediction results. For thicker coatings, elemental spectral peaks such as Ni Kα, Cr Kα, Co Kα, W Lα, and Ti Kα will or have reached the maximum information depth, and the changes in thickness have little effect on XRF intensity values. Hence, the deviation of the prediction results is large.

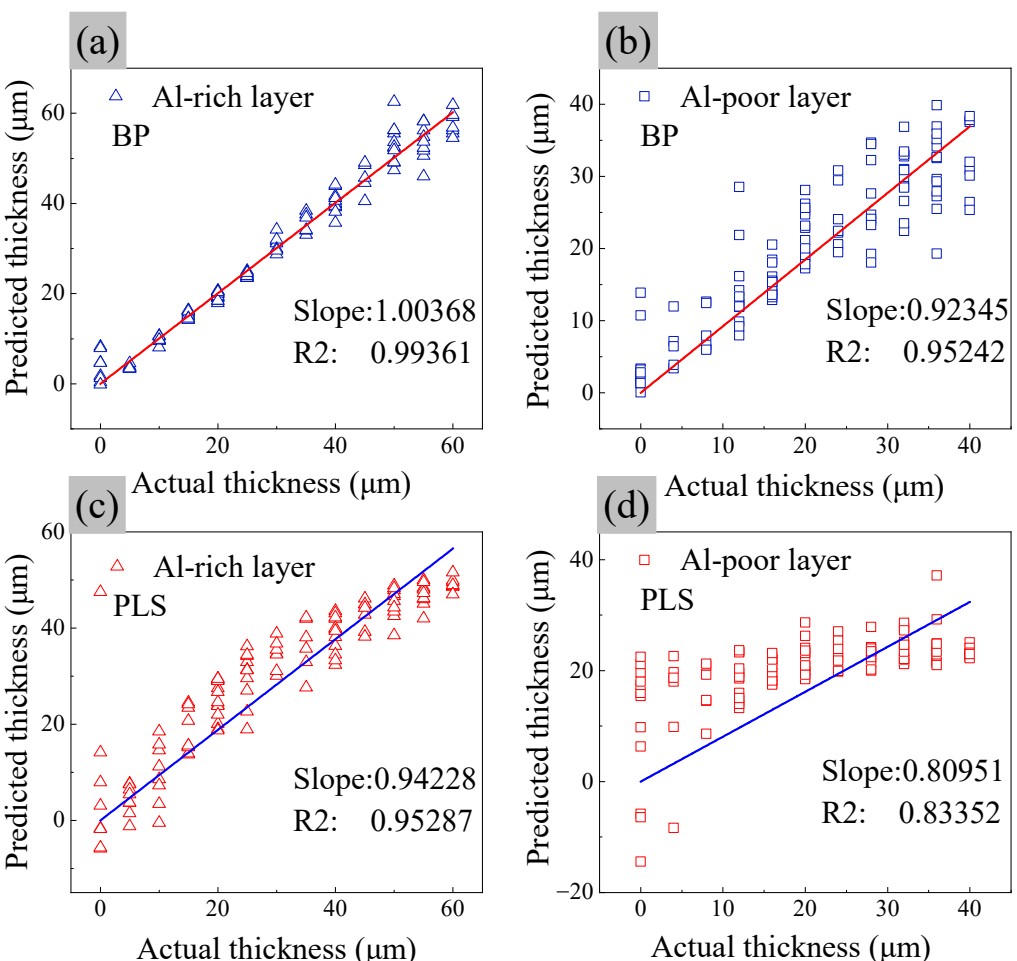

**Figure 6.** Coating thickness prediction results. (**a**) Prediction results of the BPNN model for Al-rich layer thickness; (**b**) prediction results of the BPNN model for Al-poor layer thickness; (**c**) prediction results of the PLS model for Al-rich layer thickness; (**d**) prediction results of the PLS model for Al-poor layer thickness.

Comparing the predicted results of the Al-rich layer and Al-poor layer thickness, the deviation of the predicted results of the Al-rich layer is significantly smaller than that of the Al-poor layer. Comparing the prediction results of the two models, we find that the model based on BPNN is significantly better than PLS, and the deviation of BPNN for Al-poor layer thickness prediction is similar to that of PLS for Al-rich layer thickness prediction. The PLS prediction of Al-poor layer thickness even has a negative value. The main reason for this situation is probably the nonlinear variation of the elemental XRF intensity mismatched with the linear model established by PLS. With the increase in coating thickness, the XRF of some elements reaches the maximum information depth, while the remaining elements do not, forming a nonlinear mapping overall. However, the final equation obtained in the PLS model is still a multiple linear regression model no matter what kind of transformation. Only the weights and thresholds change. The BPNN has the support of activation function, and it becomes comfortable to deal with nonlinear problems. Therefore, the BPNN-based coating thickness prediction model with higher accuracy was used in the validation process.

### 3.4. Experimental Validation

The surface coating thickness of turbine blade slices was calibrated using SEM, and XRF spectra were tested using an EDXRF instrument. The obtained XRF intensity was substituted into the BPNN model established in the previous section to predict the coating

thickness and compared with the calibrated thickness to verify the accuracy and reliability of the model.

SEM calibrations were performed for different positions of the seven slices separately, and seven sets of coating thickness values were obtained, as shown in Table 4. The comparison of the substrate's measured spectra (slice cross-section) and the coating (blade surface) is shown in Figure 7. For the XRF spectra of the coatings, the count rates of the main elements are lower than those of the substrate, except for the Ni Kα spectral line, which has a higher count rate. The results obtained from the BPNN model predictions are shown in Figure 8. For the Al-rich layer thickness (Figure 8a), the deviation between the predicted and calibrated values is small, and the average relative error was calculated to be 4.45%. For the Al-poor layer thickness (Figure 8b), the predicted deviation is larger than the Al-rich layer, consistent with the situation at the time of model establishment and mainly due to the error in propagation effects. The relative error was calculated at 16.89% and influenced by sample #7. Overall, the model was experimentally validated to achieve relatively high accuracy.

**Table 4.** Coating thickness calibration value (μm).

| Sample | 1 | 2 | 3 | 4 | 5 | 6 | 7 |
|---|---|---|---|---|---|---|---|
| Al-rich layer | 22.6 | 31.15 | 34.39 | 36.07 | 36.89 | 42.81 | 47.27 |
| Al-poor layer | 12.32 | 13.93 | 21.31 | 15.85 | 13.66 | 20.06 | 23.22 |

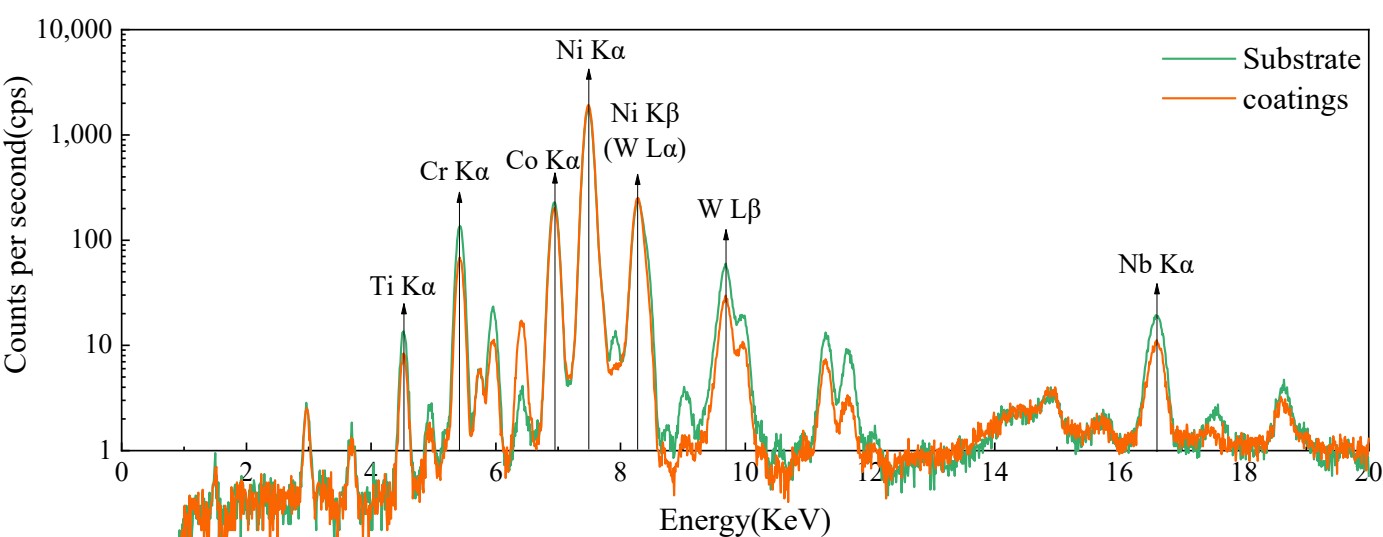

**Figure 7.** Spectral comparison of coating and substrate material.

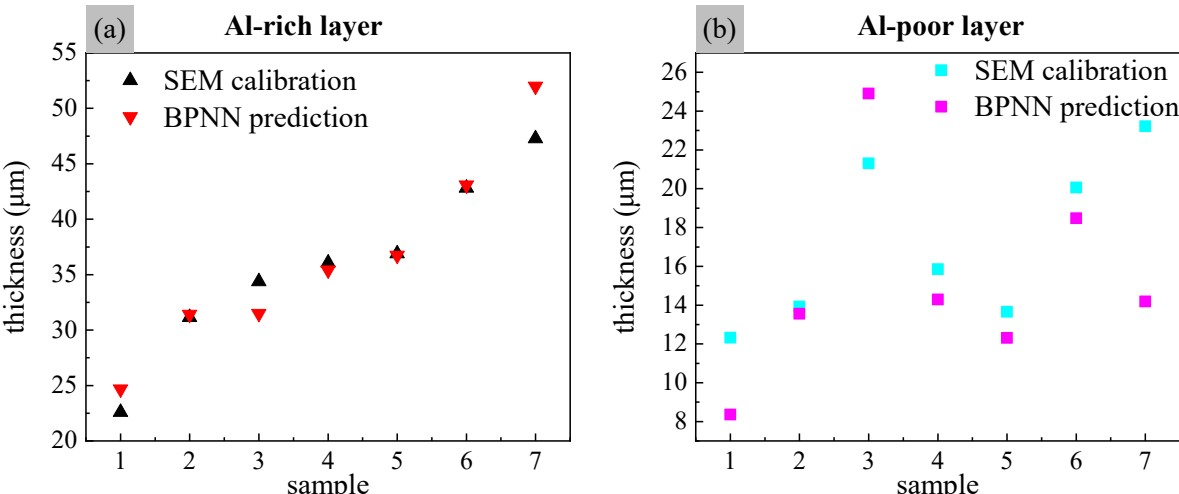

**Figure 8.** Comparison of calibrated and predicted values. (**a**) Al-rich layer thickness comparison; (**b**) Al-poor layer thickness comparison.

## 4. Conclusions

In the present paper, we have simulated the reference samples spectra using the Monte Carlo method and developed a model for predicting the thickness of the aluminized layer on the turbine blade surface using a chemometric approach. Preliminary experimental validation has shown that the method can predict coating thicknesses with relatively high accuracy. The main conclusions of this paper are as follows:

1.  The layering of aluminized layers. EDS analysis reveals that the aluminized layer on the in-service turbine blade surface can be divided into three layers with relatively stable elemental content: an Al-rich layer, an Al-poor layer, and an IDZ layer.
2.  Variation of XRF intensity for the different elements with coating thickness. The results of Monte Carlo simulations show that as the coating thickness increases, the spectral lines with lower XRF energy gradually reached the maximum information depth, while the spectral lines with higher XRF energy did not.
3.  The BNPP model is superior to the PLS model. In terms of the consistency between the predicted and actual thickness of the coating from the mathematical model, the BPNN model has the best consistency with a slope of 1.00368 and an $R^2$ of 0.99361 for the Al-rich layer, and the PLS model has a slope of 0.94228 and an $R^2$ of 0.95287. For the Al-poor layer, the BPNN model has a slope of 0.92345 and $R^2$ of 0.95242, and the PLS model has a slope of 0.80951 and $R^2$ of 0.83352. Comparing the coating thickness, prediction models show that the nonlinear BPNN model is superior to the linear PLS model.
4.  The Al-rich layer thickness prediction accuracy is higher than that of the Al-poor layer. The average relative errors of the BPNN model to predict the actual blade coating thicknesses were 4.45% for the Al-rich layer and 16.89% for the Al-poor layer, respectively. According to the prediction results, the prediction accuracy of the Al-rich layer is significantly higher than that of the Al-poor layer for both simulated and measured samples.

The present study results make it feasible to establish a prediction model for the aluminized coating thickness of turbine blades using the spectra obtained from the Monte Carlo simulation of reference samples. Further improvement in prediction accuracy may be achieved in subsequent studies by combining the evolutionary laws among the layers. In addition, for materials with a large variety of elements, complex coating systems, and difficult to fabricate reference samples, the Monte Carlo simulation method may be utilized.

**Author Contributions:** Conceptualization, Z.L. and C.W.; methodology, Z.L., C.W. and H.J.; software, Z.L. and X.L.; validation, Z.L., J.Y. and Y.Q.; formal analysis, C.W.; investigation, H.J.; resources, Z.L.; data curation, Z.L., X.L. and H.J.; writing—original draft preparation, Z.L.; writing—review and editing, Z.L. and C.W.; visualization, Z.L.; supervision, C.W.; project administration, C.W.; funding acquisition, C.W. All authors have read and agreed to the published version of the manuscript.

**Funding:** This research was funded by the National Natural Science Foundation of China, grant number 92060202, and the Innovation Fund of Air Force Engineering University, grant number CXJ2021115.

**Institutional Review Board Statement:** Not applicable.

**Informed Consent Statement:** Not applicable.

**Data Availability Statement:** Not applicable.

**Acknowledgments:** We are grateful to Xiangyang Hangtai Power Machine Factory for providing the turbine blades.

**Conflicts of Interest:** The authors declare no conflict of interest.

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
