# Peer review of "Prediction Model of Aluminized Coating Thicknesses Based on Monte Carlo Simulation by X-ray Fluorescence"

_coatings, doi:10.3390/coatings12060764_

Round 1
Reviewer 1 Report
Zhuoyue Li et al., elaborated the paper with the title “Prediction Model of Aluminized Coating Thickness Based on 2 Monte Carlo Simulation by X-Ray Fluorescence”, where compressive work regarding non-destructive studies of aluminized coatings by X-ray fluorescence technique assisted by nonlinear back-propagation neural network as alternative to fundamental parameters and Monte Carlo Simulations were presented. The presented interesting findings and standardless calibration procedures could potentially have a high impact for resolving concrete issues common in aerospace industry. Overall, I find the article well-structured followed by sufficient data, while authors should address several changes before publishing:
- English should be rigorously revised taking into consideration the grammar (e.g., missing pronouns, verbs etc) and punctuation aspects; Example: “but once the blade is damaged can not be reused”, “and it is impossible to comprehensive inspection”.
-Authors should state clearly was is the novelty of the proposed research;
- For procedure classification of the XRF methods, namely absorption, emission and relative, the authors should consider referring to the following paper that presents them in a well concise manner: doi.org/10.1016/j.surfcoat.2011.03.049;
-There is a small unbalance in the introduction chapter. The authors should consider introducing the theoretical aspects of the applied protocols in the Methods and Material section, introduction chapter should remain for presenting to the reader the studied topic, the limitations, current issues, R&D trends etc. Therefore, more relevant information to the introduction should be added (e.g., issues reported by others on NDT method for the Al coating on turbine blade surfaces);
-Also, methods and materials chapter contains information that is more suitable to be included in the introduction section;
- Taking into consideration the journal requirements and paper template, please include the manufacturer of DZ22B nickel-based high-temperature alloy / or composition; this point is applicable for all commercially available instruments applied in the paper;
-Figure 1 diagram should contain also the diagram of BPNN and PLS;
- The authors should add more information on how did the final number of data set resulted (572).
- Also, beside the detection area on the sample surface, the active detection area of the SDD detector should be mentioned;
-explanatory XRF setup diagram should help viewer visualize the simulated geometry;
- Since SEM thickness validation is commonly addressed in cross-section overview done on coatings deposited on Si substrates to ensure a minimal impact on coating, authors should explain how the blade separation process followed by wire cutting, sanding and polishing is not further affecting the thickness results;
-EDX working parameters should be mentioned;
- In section 3.3.3, there is the following statement: “the XRF intensity values of the elements will or have reached the maximum information depth”; Authors should comment regarding the saturation thickness, in relation to L and K lines;
-In figure 7, channel to keV conversion will be more suggestive; Only as a suggestion, Y axis should be logarithmic; There is a small peak between Cr and Co, it could be Fe or an escape peak? Please comment.
- What was the integration time for the spectra in figure 7.
- Since the simulations and the XRF measurements were based on a tungsten anode excitation source, how does this interfere with the W from the coating? Authors should comment.
- Conclusions should include a more solid statement, as if the NN model is applicable in validating the XRF technique;
Author Response
Special thanks to you for your good comments.
All these comments are valuable and helpful in revising and improving our paper, and they are also essential guides for our research. We have carefully studied these comments and revised them and hope to receive your approval.
For specific modifications please see the attachment.

Reviewer 2 Report
This paper proposed a prediction model of aluminized coating thickness based on Monte Carlo simulation by X-ray fluorescence. The present paper is interesting, however, to be accepted for publication the following comments need to be addressed
- Minor English changes are required in the revised manuscript
Abstract
- In the abstract, the aim of this paper is clear. The prediction errors should be presented here in numbers, to what content the back-propagation neural networks were accurate in this work?
Introduction
- The introduction section is well written, and the gaps that exist in extant literature which needs for this work are well illustrated.
Materials and Methods
- The authors should add the procedures of sample preparations for microstructure analysis by SEM.
- The specifications (model, company, city, country) of the used tools, equipment, and software should be added. (According to the guide for authors of this journal).
Results
- Figures 2b and 7, need to be modified, the Y-axis should be modified to Counts per second (cps).
Otherwise, the results are well organized and discussed.
Author Response
Special thanks to you for your good comments.
These comments are valuable and helpful for us to revise and improve our paper, and they are also important guidance for our research. We have carefully studied these comments and revised them and hope to receive your approval.
For specific modifications, please see the attachment.

Reviewer 3 Report
The article is well written and interesting.
However, the following issues need to be addressed before considering this manuscript for the publication:
- In the introduction the authors should highlight better the novelty and the motivation of their work.
- The authors should improve the conclusion section adding details to help a reader not familiar with the topic.
Author Response
Special thanks to you for your good comments.
These comments are valuable and helpful for us to revise and improve our paper, and they are also important guidance for our research. We have carefully studied these comments and revised them and hope to receive your approval.
Point 1: In the introduction the authors should highlight better the novelty and the motivation of their work.
Response 1: Following your suggestion, we have highlighted the motivation and novelty of this work in the introduction section. The surface coating of turbine blades is complex, and it is not easy to manufacture turbine blades with different coating thicknesses. Therefore, in this paper, the calculation model of aluminized layer thickness is established and verified by simulating different thicknesses of aluminized blades using the Monte Carlo method.
The motivation of this paper is to establish a prediction model of aluminized layer thickness based on Monte Carlo simulation for nondestructive testing of aluminized coatings on turbine blade surfaces.
The innovation of this study is mainly in the XRF Monte Carlo simulation of complex coating samples. It provides a new way of thinking for those cases where a series of reference samples are needed to establish a calibration curve, but the reference samples are difficult to manufacture.
Point 2: The authors should improve the conclusion section adding details to help a reader not familiar with the topic.
Response 2: Based on your suggestion, we have added the coating thickness prediction data in the conclusion section to make the conclusion more acceptable to readers. The details are as follows:
In terms of the consistency between the predicted and actual thickness of the coating from the mathematical model, the BPNN model has the best consistency with a slope of 1.00368 and R2 of 0.99361 for the Al-rich layer and the PLS model has a slope of 0.94228 and R2 of 0.95287. For the Al-poor layer, the BPNN model has a slope of 0.92345 and R2 of 0.95242, and the PLS model has a slope of 0.80951 and R2 of 0.83352. Comparing the coating thickness prediction models shows that the nonlinear BPNN model is superior to the linear PLS model. The average relative errors of the BPNN model to predict the actual blade coating thicknesses were 4.45% for the Al-rich layer and 16.89% for the Al-poor layer, respectively. According to the prediction results, the prediction accuracy of the Al-rich layer is significantly higher than that of the Al-poor layer for both simulated and measured samples.
Special thanks to you for your good comments.